# A Laboratory-Developed Assay for the Simultaneous Detection of *Aspergillus fumigatus* and *Pneumocystis jirovecii* Pulmonary Pathogens

**DOI:** 10.3390/jof11040280

**Published:** 2025-04-02

**Authors:** Margherita Cacaci, Debora Talamonti, Giulia Menchinelli, Damiano Squitieri, Riccardo Torelli, Elena De Carolis, Giulia De Angelis, Maurizio Sanguinetti, Brunella Posteraro

**Affiliations:** 1Dipartimento di Scienze Biotecnologiche di Base, Cliniche Intensivologiche e Perioperatorie, Università Cattolica del Sacro Cuore, Largo A. Gemelli 8, 00168 Rome, Italy; margherita.cacaci@unicatt.it (M.C.); damiano.squitieri@unicatt.it (D.S.); giulia.deangelis@unicatt.it (G.D.A.); brunella.posteraro@unicatt.it (B.P.); 2Dipartimento di Scienze di Laboratorio ed Ematologiche, Fondazione Policlinico Universitario A. Gemelli IRCCS, Largo A. Gemelli 8, 00168 Rome, Italy; debora.talamonti@gmail.com (D.T.); giulia.menchinelli@policlinicogemelli.it (G.M.); riccardo.torelli@policlinicogemelli.it (R.T.); elena.decarolis@policlinicogemelli.it (E.D.C.); 3Unità Operativa “Medicina di Precisione in Microbiologia Clinica”, Direzione Scientifica, Fondazione Policlinico Universitario A. Gemelli IRCCS, Largo A. Gemelli 8, 00168 Rome, Italy

**Keywords:** pulmonary fungal infection, laboratory-developed assay, molecular diagnostics, Open Access, Panther Fusion

## Abstract

Invasive fungal diseases are a significant threat in immunocompromised patients, underscoring the need for rapid and accurate diagnostics. This study describes the development and validation of a real-time PCR-based laboratory-developed assay (LDA) on the Panther Fusion system for the simultaneous detection of *Aspergillus fumigatus* (AF) and *Pneumocystis jirovecii* (PJ) in bronchoalveolar lavage fluid (BALF) samples. The assay was evaluated using 239 clinical BALF samples, including cases confirmed positive for AF or PJ by reference mycological methods. Rigorous optimization ensured compatibility with the automated workflow of the Panther Fusion system, which addresses challenges such as BALF viscosity and fungal DNA recovery. No cross-reactivity with non-target fungal species was observed, and the assay demonstrated high analytical sensitivity and specificity. Only two false-negative results were reported, which could plausibly be reclassified as true negatives when interpreted alongside the serum beta-d-glucan and galactomannan assay results. For PJ detection, the assay showed excellent concordance with the OLM PneumID assay, supporting its reliability in clinical settings. The dual-target approach facilitates the simultaneous detection of both pathogens within a single workflow, improving diagnostic efficiency. The AF/PJ LDA represents a robust and scalable alternative to existing molecular assays, with the potential to enhance routine diagnostics for pulmonary fungal infections.

## 1. Introduction

Pulmonary fungal infections caused by *Aspergillus fumigatus* and *Pneumocystis jirovecii* present significant diagnostic challenges in immunocompromised patients, such as those undergoing organ transplantation or receiving immunosuppressive therapy [1,2]. *A. fumigatus*, a ubiquitous mold, is responsible for invasive pulmonary aspergillosis, while *P. jirovecii* causes *Pneumocystis* pneumonia (PcP), a life-threatening infection that is especially common in patients with HIV/AIDS or hematologic malignancies.

Conventional culture-based diagnostic methods for fungal pathogens face several limitations, including low sensitivity and prolonged incubation periods. Notably, *P. jirovecii* cannot be cultivated in vitro, making molecular-based diagnostics essential [3]. In response to these diagnostic gaps, laboratory-developed tests/assays (LDTs/LDAs) have emerged as valuable tools, offering rapid and accurate pathogen detection [4]. The fully automated Panther Fusion system (Hologic, San Diego, CA, USA) for nucleic acid extraction and polymerase chain reaction (PCR), with its Open Access functionality, enables the adaptation of molecular-based assays for various targets, including influenza A virus subtyping [5], tickborne diseases [6], measles [7], clade II human mpox virus [8], and macrolide-associated mutations in *Mycoplasma genitalium* [9], alongside in vitro diagnostic (IVD) assays [10].

In line with these efforts to leverage existing molecular infrastructures, such as SARS-CoV-2 testing platforms [11], to detect other critical pathogens, our assay builds on this approach by adapting the Panther Fusion system for the dual detection of *A. fumigatus* and *P. jirovecii*. These pathogens have been included in the WHO fungal pathogen priority list due to their clinical significance, with *A. fumigatus* classified as “critical priority” and *P. jirovecii* as “medium priority” [12].

This study describes the development and validation of an LDA using real-time PCR on the Panther Fusion system, marking its first application for the detection of fungal pathogens. Unlike previous implementations for viral and bacterial targets, this assay was specifically designed to simultaneously detect *A. fumigatus* and *P. jirovecii* in bronchoalveolar lavage fluid (BALF) samples. While the assay provides a fully automated and streamlined workflow for the rapid and sensitive detection of these fungal pathogens, some limitations remain, as demonstrated by the occurrence of false-negative results during validation.

## 2. Materials and Methods

### 2.1. Study Samples

The samples used in this study included both primary (*n* = 239) and contrived (*n* = 28) clinical BALF samples. The primary samples were obtained from patients referred to the clinical microbiology laboratory of a large tertiary care hospital in Rome, Italy, between June 2022 and December 2023, for mycological examination as part of routine diagnostic care. The diagnostic workup included a fungal culture on Sabouraud dextrose agar plates and/or a galactomannan (GM) assay [13] for suspected pulmonary aspergillosis, and the PneumID (OLM Diagnostics, Newcastle-upon-Tyne, UK) real-time PCR assay [14] and/or serum 1,3-β-d-glucan (BDG) assay [15] for suspected PcP, in accordance with the EORTC/MSGERC-defined criteria for mycological evidence of invasive fungal disease (IFD) [16].

The primary BALF samples with positive or negative results for *A. fumigatus* and/or *P. jirovecii* were included. The *A. fumigatus*-positive BALF results were defined as the growth of fungal colonies in culture, alone or in combination with a GM level ≥ 1.0 cutoff [17]. The *P. jirovecii*-positive BALF results were defined as positive PneumID real-time PCR results (≤35.0 cycles), alone or in combination with a BDG serum concentration ≥ 80 pg/mL [14]. Additionally, 104 samples tested positive for bacterial species, including *Pseudomonas aeruginosa* (21 samples), *Staphylococcus aureus* (15 samples), *Klebsiella pneumoniae* (9 samples), *Acinetobacter baumannii* (6 samples), and *Haemophilus influenzae* (5 samples), among others. Of these, 10 samples were concomitantly positive for the targeted pathogens. Viruses were identified in 26 samples, with SARS-CoV-2 detected in 8 samples, cytomegalovirus in 7, rhinovirus/enterovirus in 4, influenza virus A in 3, Epstein-Barr virus in 3, and bocavirus in 1. Among these, 11 samples were concomitantly positive for the targeted pathogens. No samples tested positive for *Mucorales*, *Scedosporium*, or *Histoplasma*. The BALF samples that were negative for fungal or bacterial species, including the targeted pathogens, were used to prepare the contrived BALF samples by spiking known concentrations of non-targeted microbial species (1 per sample), including bacterial (*n* = 19), filamentous fungal (*n* = 7), and yeast (*n* = 2) species, as detailed in Appendix A.

Aliquots of the BALF samples were stored at −20 °C after the initial testing and used for subsequent analyses. Both the primary and contrived BALF samples were tested using the Hologic Panther Fusion system to support the development and validation of the LDA.

In addition, two types of DNA plasmids containing *A. fumigatus*- or *P. jirovecii*-specific targets, obtained as described below, were used as serial dilutions during the optimization and performance assessment of the LDA.

### 2.2. Panther Fusion-Based LDA

#### 2.2.1. Primers and Probes

The primers and probes for the *A. fumigatus* (AF)/*P. jirovecii* (PJ) real-time PCR assay, designed for use on the Panther Fusion system, were synthesized by Eurofins Scientific (Rome, Italy) to amplify the target regions within the fungal genes encoding the ribosomal RNA internal transcribed spacer 1 (ITS1) [4] and dihydrofolate reductase (DHFR) [18]. The sequences, previously published, were as follows: (i) AF-PCR: forward primer, 5′-GCCCGCCGTTTCGAC-3′; reverse primer, 5′-CCGTTGTTGAAAGTTTTAACTGATTAC-3′; probe, 5′-HEX-CCCGCCGAAGACCCCAACATG-BHQ1-3′ [19] and (ii) PJ-PCR: forward primer, 5′-GGCTGATCAAAGAAGCATGGATA-3′; reverse primer, 5′-CGGCATAGACATATTCGATACTTGTT-3′; probe, 5′-ROX-TGCGTGAAACAGATACATGGAGCTCTACCC-BHQ2-3′ [20].

The Basic Local Alignment Search Tool (BLAST) program from the National Center for Biotechnology Information (https://blast.ncbi.nlm.nih.gov/Blast.cgi); accessed on 28 February 2025) was used to verify the primer specificity, while the OligoAnalyzer tool (https://www.idtdna.com; accessed on 28 February 2025) was applied to detect potential primer dimers, which are known to reduce PCR efficiency in probe-based assays [21]. Since the LDA was designed as a multiplex assay to simultaneously detect both AF and PJ targets, primer pair compatibility was tested to exclude any potential cross-reactivity within the primer–probe reconstitution (PPR) mix.

#### 2.2.2. Target Plasmid DNA

Genomic DNA from the *A. fumigatus* ATCC 13093 reference strain was used to amplify the ITS1 gene region [22] by PCR using a BioRad CFX96 thermal cycler (BioRad, Hercules, CA, USA) and the KAPA HiFi PCR mix (Kapa Biosystems, Wilmington, MA, USA). The primers (forward: 5′-AACGACTCCCCAGAGCCGGAAAG-3′; reverse: 5′-CTCGCCGTTACTGAGGCAATCCC-3′) were designed to encompass the LDA-target ITS1 region (GenBank accession number: GU319985.1). The PCR product was purified and sequenced using the SeqStudio Genetic Analyzer (Thermo Fisher Scientific, Waltham, MA, USA) to confirm the sequence identity before being cloned into a pCR 2.1 vector (Zero Blunt PCR Cloning Kit, Thermo Fisher Scientific), as previously described [20].

A previously constructed DNA plasmid carrying the LDA target *P. jirovecii* DHFR gene region [20] was recovered from the laboratory’s frozen stock and used in this study.

Before use, the plasmids containing the AF-ITS1 and PJ-DHFR LDA targets, as well as the genomic DNAs used for the BALF sample spiking, were quantified using a NanoDrop One spectrophotometer (Thermo Fisher Scientific). The number of DNA copies per LDA reaction was then determined.

#### 2.2.3. Optimization and Performance Experiments

Experiments were performed to optimize the final concentrations of the real-time PCR reagents (all provided by Hologic, except for the nuclease-free water, which was provided by Thermo Fisher Scientific). Various concentrations for the MgCl_2_ (2, 3, and 4 mM), KCl (50 and 65 mM), the primers (0.2 and 0.4 µM), and the probes (0.1 and 0.2 µM) were tested in the PPR mix, as detailed in Appendix A. The final concentrations for the optimized PPR mix were based on those that produced the lowest cycle threshold (Ct) values and the greatest Ct precision.

To perform the PCR, the PPR mix was overlaid with the Panther Fusion oil reagent (Hologic) and loaded onto the Panther Fusion instrument, which rehydrated the single-reaction lyophilized enzyme and nucleotide mixture with the PPR mix to create the PCR master mix. A total of 20 µL of this master mix was combined with an eluted DNA sample (detailed below), of which 5 µL was subjected to amplification, which consisted of 1 cycle at 95 °C for 2 min followed by 40 cycles at 95 °C for 5 s, 55 °C for 13 s, and 60 °C for 19 s.

To determine the limit of detection (LoD) for the AF/PJ-LDA, a 10- or 4-fold dilution series of the AF/PJ DNA plasmid was prepared using a microbiologically negative BALF sample as the diluent. A seven-point standard curve was constructed using five replicates per DNA plasmid dilution, and the dilutions were also tested for precision analysis.

The PCR efficiency was calculated from the slope of the linear portion of the log-transformed standard curve, according to the equation e = 10^−1/slope^, where “e” represents the theoretical efficiency and “slope” is derived from a plot of the logarithm of the initial DNA concentration (*x*-axis) against the Ct value (*y*-axis) [21].

The analytical specificity was assessed by testing the contrived BALF samples (Appendix A), where negative results for both AF/PJ LDA targets were expected.

#### 2.2.4. BALF Sample Testing

Before testing, 500 μL of the BALF sample was pretreated for 30 min at 37 °C with 500 μL of Sputasol (Thermo Fisher Scientific), a dithiothreitol-containing mucolytic reagent that reduces the viscosity of lower respiratory tract specimens. Then, 500 μL of the mixture was transferred to a Panther Fusion Specimen Lysis Tube. Alternatively, 250 μL of the DNA-containing sample, prepared for the LDA optimization/performance experiments, was combined with 250 μL of the pretreated BALF sample in a Panther Fusion Specimen Lysis Tube.

Automated nucleic acid extraction was performed on 300 μL of the processed sample using Panther Fusion Extraction Reagent-S, which contained Hologic’s Internal Control DNA. The extracted total nucleic acid was eluted in 50 μL, of which 5 μL was amplified, resulting in an effective sample volume of 15 μL. Before running the PCR on the Panther Fusion instrument (Hologic), a PCR master mix, created as described above, was added. The AF analyte was detected in the HEX channel and the PJ analyte in the ROX channel. The PCR parameters, including thermal cycling conditions (as detailed above) and setting the fluorophore channel’s maximum valid Ct to 40, were established using Hologic Open Access software (version 2.1.2.1).

### 2.3. Statistical Analysis

The data were presented as numbers with percentages, as means ± standard deviation (SD), or as coefficients of variation (CV), as appropriate. Linear regression analysis was conducted to determine the correlation coefficients (R^2^) using GraphPad Prism version 10.4.0. Probit analysis was conducted to determine the LoD, defined as the lowest concentration at which 95% of the LDA-positive samples were detected, using RStudio version 2023.12.1.

According to the specified criteria for IVD assays [23], the acceptable accuracy was defined as ≥90% agreement between the reference method and the LDA, while the acceptable precision was defined as ≥90% agreement within ±3 Ct values among LDA replicates. True positive and true negative BALF samples, as classified by the reference method, were used to calculate the positive percent agreement (PPA) and negative percent agreement (NPA), along with the respective 95% confidence intervals (CIs), between the LDA and reference method results.

## 3. Results

Initial experiments were conducted to optimize the PPR mix for *A. fumigatus* (AF) and *P. jirovecii* (PJ). Appendix A details the reagent concentrations tested, including MgCl_2_, KCl, primers, and probes, evaluated in a stepwise manner. The optimized concentrations shown in Table 1 provided the best conditions for reliable amplification.

The amplification curves obtained with these optimized conditions, presented in Appendix A, demonstrate consistent improvements in Ct values and precision, confirming the suitability of the finalized PPR mix for the simultaneous detection of AF and PJ.

The analytical sensitivity (i.e., LoD) for the AF/PJ LDA was determined using a dilution series of DNA plasmids (Table 2). For AF, the assay consistently detected 1.30 log_10_ copies/reaction in all replicates (5/5) with a mean Ct value of 34.5, standard deviation (SD) of 0.55, and a coefficient of variation (CV) of 1.60%. The detection dropped to 2/5 replicates at 0.70 log_10_ copies/reaction. Similarly, for PJ, the assay detected 1.00 log_10_ copies/reaction in all replicates (5/5), with a mean Ct value of 34.7, SD of 0.90, and a CV of 2.59%. Partial detection was observed at 0.39 log_10_ copies/reaction, where only 2/5 replicates were positive. The LoD was determined to be 1.25 log_10_ copies/reaction for the AF target and 1.02 log_10_ copies/reaction for the PJ target.

Linear regression analysis of the Ct values across the dilution series demonstrated a strong correlation with the expected DNA concentrations, with R² values of 0.98 for both AF and PJ (Figure 1). The calculated PCR efficiency was 95.1% for AF and 93.9% for PJ, confirming the robustness of the assay under optimized conditions.

The analytical specificity for the AF/PJ LDA was determined using a panel of BALF samples, each spiked with DNA from 28 bacterial and fungal species other than the two pathogens included in the LDA (Appendix A). None of these samples yielded positive detection results, demonstrating a lack of cross-reactivity.

The AF/PJ LDA was evaluated using 239 clinical BALF samples, stratified according to standard care testing results for *A. fumigatus* and *P. jirovecii* pulmonary pathogens (Table 3). Among the 22 samples positive by AF-LDA, all were correctly identified as positive for the AF culture, with no false positives reported. However, one sample among the AF-negative group (*n* = 216) yielded a false-negative result at the AF-LDA. This sample tested negative by the GM assay but was identified as PJ-positive according to the PJ-LDA, PneumID assay, and BDG assay results.

For PJ detection, 11 samples were positive by the PJ-LDA, and all were confirmed as positive by the PneumID assay. However, one false-negative result was observed in a PJ-negative sample group (*n* = 227). This sample had a Ct value near the positivity threshold in the PneumID assay, and the corresponding patient tested negative for BDG in the serum sample.

The combined detection of AF and PJ was observed in one BALF sample, which was accurately identified by the AF/PJ LDA. Overall, the LDA demonstrated high concordance with standard care diagnostic methods, supporting its reliability for the simultaneous detection of AF and PJ in clinical BALF samples.

The diagnostic performance of the AF/PJ LDA was evaluated using 239 clinical BALF samples (Table 4). The assay demonstrated a PPA of 95.8% for AF and 92.3% for PJ, with an NPA of 100% for both targets. Among the positive samples, the Ct values ranged from 25.1 to 35.7 for AF and from 22.6 to 34.3 for PJ, reflecting robust detection across DNA concentrations. One false-negative result was observed for each target, as detailed in Table 3, while no false positives were reported.

## 4. Discussion

We utilized the Open Access functionality of the Hologic Panther Fusion system to develop a novel real-time PCR-based assay in our laboratory for the simultaneous detection of two clinically significant fungal pathogens, *A. fumigatus* (AF) and *P. jirovecii* (PJ), in BALF samples. Rigorous optimization and validation experiments ensured that the AF/PJ LDA met the performance specifications required for IVD assays, demonstrating excellent analytical and diagnostic sensitivity and specificity. Importantly, no cross-reactivity with non-AF/PJ fungal pathogens or false-positive detections were observed during the BALF sample testing. Only two samples (0.8%)—one positive for AF and the other for PJ as confirmed by reference mycological assays—were classified as false negatives by the LDA.

Over the past decade, molecular diagnostics have been extensively explored for IFD, particularly in relation to pulmonary aspergillosis and PcP [3,24]. The incorporation of PCR as a mycological criterion for IFD diagnosis has significantly enhanced diagnostic accuracy, especially in at-risk immunocompromised populations [16]. However, the field has also witnessed the emergence and subsequent discontinuation of several commercial assays, particularly for AF, highlighting the challenges in ensuring long-term assay reliability and clinical utility [4,25]. Against this backdrop, the AF/PJ LDA presents a novel approach by combining two fungal targets, AF and PJ, into a multiplex assay. This strategy aligns with the growing demand for syndromic testing panels, such as the BIOFIRE FILMARRAY Pneumonia and Pneumonia Plus Panels (bioMérieux, Marcy l’Étoile, France), which include multiple bacterial and viral targets but lack fungal coverage.

Using more than one single-target assay to detect specific genera or species can be costly in terms of labor and turnaround time, particularly in clinical microbiology laboratories that handle a high volume of diversified mycological requests daily. Similarly to our approach, a duplex real-time PCR assay, such as MycoGENIE Real-Time PCR (Ademtech, Pessac, France), has been developed to detect *Aspergillus* DNA and *Mucorales* DNA by targeting the 28S ribosomal RNA gene region [3]. Both *Aspergillus* spp. and *Mucorales* are significant causes of pulmonary infections, particularly in immunocompromised patients, often leading to severe IFD. In a retrospective study evaluating the diagnostic performance of serum biomarkers for invasive aspergillosis in hematology patients, MycoGENIE PCR successfully identified two probable co-infections with *Aspergillus* spp. and *Mucorales* [26]. While further advancements are anticipated to extend the AF/PJ LDA into a triple-target assay as soon as possible by including *Mucorales*, it is notable that one BALF sample in our study tested positive for both AF and PJ. This highlights the importance of developing multiplexed assays capable of detecting multiple significant fungal pathogens causing respiratory infections.

Unlike assays targeting AF, which can utilize samples such as blood, serum, or plasma, BALF remains the preferred sample for PJ detection in most commercially available assays [3]. It is therefore unsurprising that the OLM PneumID—the comparator assay in this study—has primarily been validated (CE-IVD) for BALF as a clinical sample. In the Fungal PCR Initiative (FPCRI) evaluation conducted by Gits-Muselli et al. [18], PneumID demonstrated high analytical sensitivity, ranking within the top five of the 19 PJ-PCR assays evaluated. Similarly, a subsequent performance evaluation by Price et al. [14] highlighted its excellent sensitivity and specificity, particularly for respiratory samples.

In our study, all but one BALF sample with positive PneumID assay results were concordant with the PJ-LDA. Considering Ct values as an indicator of fungal burden, in conjunction with serum BDG positivity to support a probable PcP diagnosis [16], the false-negative PJ-LDA result could plausibly be reclassified as a true negative. This is because the Ct value for the PneumID assay in this sample exceeded the established false-positivity threshold of <33.1 [14], and the patient’s concomitant serum BDG assay result was negative. Interestingly, all of the PJ-LDA positive samples were from patients with concomitantly high serum BDG levels (>500 pg/mL). Conversely, the patients with BALF samples that tested negative for PJ-LDA and had available serum BDG results showed BDG levels below 80 pg/mL, a value widely recognized as the positivity threshold for IFD in the Fungitell BDG assay [16]. Another sample, classified as a false negative for the AF-LDA based on the AF-negative culture result, tested negative in the GM assay but positive in the PJ-LDA, PneumID, and BDG. Therefore, this sample could plausibly be reclassified as positive only for PJ. The concomitant negativity for GM, combined with the culture positivity, suggests the possibility of contamination, thereby justifying its exclusion as an AF-positive sample.

The development of fungal diagnostic assays poses unique challenges, particularly in sample preparation and nucleic acid extraction, which are often more complex than for bacterial or viral pathogens [25]. These difficulties have contributed to the limited inclusion of fungal targets in existing syndromic panels. Many manual and automated protocols have been developed for fungal DNA extraction, but their performance can vary significantly depending on the pathogen and specimen type [4]. For the Panther Fusion system, which incorporates an automated extraction process, this study represents the first application for fungal diagnostics. Unlike manual protocols (e.g., those used with the PneumID assay), the proprietary nature of the Panther Fusion extraction system precludes detailed insight into its mechanisms. However, by following established recommendations to reduce the BALF sample viscosity [3], we ensured optimal compatibility with the Panther Fusion workflow. This approach effectively addresses the technical barriers posed by BALF samples, particularly their viscosity and the structural complexity of fungal cell walls, while providing an efficient solution for fungal DNA recovery.

The choice of primers is a critical factor influencing the performance of PCR assays, as it directly affects their sensitivity and specificity. False-negative results can occur when primers target variable regions of fungal genomes, leading to suboptimal binding in certain organisms. In this study, the primers were carefully selected to target the ITS1 region within the ribosomal RNA gene cluster for AF and the DHFR gene for PJ. The ITS1 region is a widely recognized target for fungal diagnostics due to its high copy number and relatively conserved sequences across fungal species, making it suitable for detecting AF with high sensitivity. For PJ, the DHFR gene was chosen based on the findings from our previous study published in 2021 [20], which demonstrated its effectiveness as a reliable target for molecular detection. DHFR is particularly advantageous due to its low variability and its diagnostic relevance in identifying PJ, as it circumvents some of the challenges associated with other, more variable genetic regions. By selecting these specific targets, the AF/PJ LDA ensures robust detection while minimizing the risk of false-negative results, even in cases where genetic variability could potentially compromise assay performance.

A potential risk for molecular diagnostic assays, such as the one described in this study, is that they may be perceived as mere technological exercises without practical clinical applicability. The primary aim of this research, however, was to develop a robust assay that is seamlessly integrable into routine diagnostic workflows. Although the AF/PJ LDA demonstrated high analytical and diagnostic performance in this study, its clinical validation remains incomplete, as we lack information on how the routine mycological results were interpreted by the treating physicians based on the clinical context. This is particularly relevant for PJ, where distinguishing colonization from active infection relies heavily on clinical correlation [27], even when the PCR results, such as those from the OLM PneumID, are concordant. In this context, the need for a multidisciplinary approach to diagnosing pulmonary fungal infections should be highlighted, integrating clinical, radiological, and serological findings (e.g., BDG and GM assays) for a comprehensive assessment [16].

The AF/PJ LDA was designed to address key challenges in fungal diagnostics, including the need for rapid, multiplexed detection of *A. fumigatus* and *P. jirovecii*. While the Panther Fusion system provides an efficient workflow, its proprietary extraction process limits assay adaptability to other platforms. Nevertheless, it overcomes the complexities of current BALF sample processing workflows, which typically require multiple steps prior to PCR-based molecular testing. Future studies should focus on further validation of the assay in larger, multicenter cohorts and explore its integration into diverse laboratory platforms to maximize clinical utility.

In conclusion, the AF/PJ assay represents a promising alternative to existing molecular diagnostics, combining multiplexed detection with a streamlined workflow. Its ability to address the challenges of fungal diagnostics while offering scalability highlights its potential for integration into routine clinical practice.

## Figures and Tables

**Figure 1 jof-11-00280-f001:**
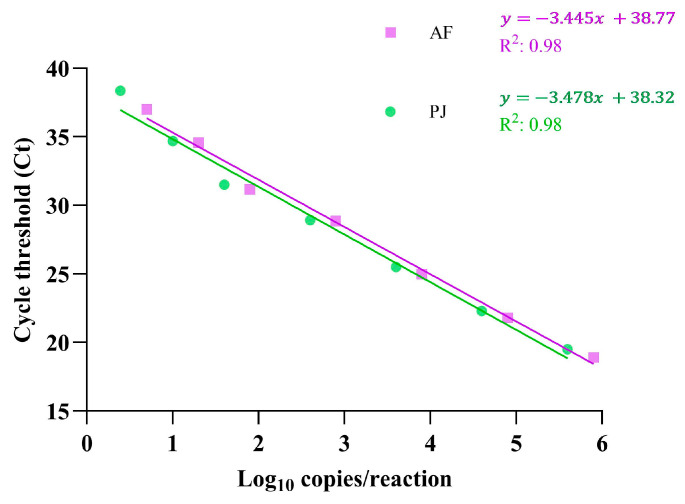
Linearity of the AF and PJ LDA determined on the Panther Fusion instrument. Each dilution of plasmid DNA spiked into BALF was tested in quintuplicate, and the mean Ct value at each DNA concentration (log_10_ copies/reaction) was plotted. Linear regression analysis was performed to evaluate the correlation between the DNA concentration and Ct values, with correlation coefficients (R²) indicating the assay’s linearity for each target. The equations representing the linear regression lines are also displayed.

**Table 1 jof-11-00280-t001:** Optimized final concentrations of PCR reagents.

Reagent	Concentrations	
	AF-LDA	PJ-LDA
MgCl_2_, mM	4	4
KCl, mM	50	50
Tris, mM	12.5	12.5
Forward primer, µM	0.2	0.2
Reverse primer, µM	0.2	0.2
Probe, µM	0.1	0.1
Internal control	0.5×	0.5×

AF-LDA, *A. fumigatus*-laboratory developed assay; PJ-LDA, *P. jirovecii*-laboratory developed assay.

**Table 2 jof-11-00280-t002:** Analytical limit of detection for the *A. fumigatus* (AF) and *P. jirovecii* (PJ) LDA.

Assay Target	Concentration (log_10_ Copies/Reaction)	No. of Replicates Detected/No. of Replicates Tested	Mean Ct	Standard Deviation	Coefficient of Variation (%) ^a^
AF					
	5.90	5/5	18.9	0.30	1.60
	4.90	5/5	2.8	0.35	1.62
	3.90	5/5	24.9	0.21	0.86
	2.90	5/5	28.8	0.32	1.13
	1.90	5/5	31.1	0.23	0.75
	1.30	5/5	34.5	0.55	1.60
	0.70	2/5	37.1	0.52	1.42
PJ					
	5.60	5/5	19.5	0.47	2.42
	4.60	5/5	22.3	0.33	1.50
	3.60	5/5	25.4	0.35	1.37
	2.60	5/5	28.9	0.43	1.49
	1.60	5/5	31.5	0.23	0.72
	1.00	5/5	34.7	0.90	2.59
	0.39	2/5	38.3	0.52	1.38

LDA, laboratory-developed assay; Ct, cycle threshold. ^a^ The coefficient of variation (CV) was calculated as: CV (%) = (standard deviation/mean Ct) × 100.

**Table 3 jof-11-00280-t003:** Standard care testing results for *A. fumigatus* and *P. jirovecii* in patients with BALF samples tested using the Panther Fusion LDA.

Panther Fusion LDA	No. of Positive (with Values) or Negative Results for Samples Tested by:							
	AF Culture (BALF Samples, *n* = 239)		GM Assay (BALF Samples, *n* = 57)		PneumID Assay (BALF Samples, *n* = 239)		BDG Assay (Serum Samples, *n* = 64)	
	Positive	Negative	≥1 Cutoff	<1 Cutoff	≤35 Ct	>35 Ct	≥80 pg/mL	<80 pg/mL
AF positives (*n* = 22)	22	0	11 (2–>10)	2	–	–	–	–
AF negatives (*n* = 216)	1 ^a^	215	9 (2–>10)	35	–	–	–	–
PJ positives (*n* = 11)	–	–	–	–	11 (17–29)	0	11 (>500)	0
PJ negatives (*n* = 227)	–	–	–	–	1 (31) ^b^	226	0	52
AF/PJ positives (*n* = 1)	1	0	–	–	1 (23)	0	1 (>500)	0

Dashes (–) indicate results not provided because they were considered inapplicable or unavailable. AF, *Aspergillus fumigatus*; PJ, *Pneumocystis jirovecii*; GM, galactomannan; BDG, 1,3-β-D-glucan; LDA, laboratory-developed assay; Ct, cycle threshold. ^a^ This sample, classified as false negative for the AF-LDA, tested negative in the GM but positive in the PJ-LDA, PneumID, and BDG. ^b^ This sample, classified as false negative for the PJ-LDA, tested positive in the PneumID with a Ct value close to the positivity threshold and was obtained from a patient whose concomitant serum sample tested negative in the BDG.

**Table 4 jof-11-00280-t004:** Results of the AF/PJ LDA for clinical BALF samples stratified by target.

Results	AF	PJ
No. matched positive	23	12
No. matched negative	215	226
No. missed	1	1
Range of Ct values	25.1–35.7	22.6–34.3
PPA (95% CI)	95.8% (78.9–99.9%)	92.3% (64.0–100%)
NPA (95% CI)	100% (98.3–100%)	100% (98.4–100%)

AF: *Aspergillus fumigatus*; PJ: *Pneumocystis jirovecii*; PPA: positive percent agreement; NPA: negative percent agreement; Ct: cycle threshold.

## Data Availability

The original contributions presented in the study are included in the article, further inquiries can be directed to the corresponding author.

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
