# Peer review of "A Laboratory-Developed Assay for the Simultaneous Detection of Aspergillus fumigatus and Pneumocystis jirovecii Pulmonary Pathogens"

_jof, 2025, doi:10.3390/jof11040280_

Round 1

Reviewer 1 Report

This is only a methodological paper.

Authors mention that “patients referred to the clinical microbiology laboratory of a large tertiary-care hospital in Rome, Italy, between June 75 2022 and December 2023, for mycological examination as part of routine diagnostic care” This is not enpugh and patients should be better characterized: bacterial infections and which ones? Viral infections and which ones? Controls?

Very few positive samples are seen : less than 10 % of the total samples. This is quite unbalanced vs the "controls".

No test for other putative fungal co infections: Mucorales, scedosporium, histoplasma…..

This ms is not really exciting and it is not the first time that PCR is used to diagnose these two fungal infections.

Author Response

Comments 1: This is only a methodological paper.

Response 1: We acknowledge that our study primarily focuses on the methodological development and validation of the AF/PJ LDA. However, we believe that the clinical applicability of the assay is an essential aspect of our work. The primary objective of this study was to develop a molecular assay designed for seamless integration into routine clinical diagnostics, addressing specific challenges in fungal detection. While our manuscript focuses on analytical performance, we have discussed its potential impact in a diagnostic setting and its advantages over existing methodologies.

Comments 2: Authors mention that “patients referred to the clinical microbiology laboratory of a large tertiary-care hospital in Rome, Italy, between June 2022 and December 2023, for mycological examination as part of routine diagnostic care” This is not enough, and patients should be better characterized: bacterial infections and which ones? Viral infections and which ones? Controls?

Response 2: We acknowledge the reviewer’s concern regarding the characterization of the patient cohort. While our study is primarily methodological, we recognize the importance of providing additional context on the clinical background of the patients. To address this, we have included further details in the Materials and Methods section, specifying the presence of concurrent bacterial or viral infections where applicable. Regarding controls, we clarify that our study was not designed as a case-control investigation in the strictest sense (i.e., comparing infected vs. non-infected patients). Instead, given its methodological nature, we included BALF samples that were microbiologically negative for A. fumigatus and P. jirovecii as a reference for comparison, rather than true “controls” in a clinical case-control framework.

Comments 3: Very few positive samples are seen less than 10 % of the total samples. This is quite unbalanced vs the “controls”.

Response 3: The distribution of positive and negative samples in our study reflects the actual epidemiology of A. fumigatus and P. jirovecii infections in the clinical setting where the study was conducted. The relatively low proportion of positive samples is consistent with the expected prevalence of these pathogens in routine diagnostic workflows. Despite this imbalance, the number of positive cases remains sufficient to assess the analytical performance of the AF/PJ LDA.

Comments 4: No test for other putative fungal co infections: Mucorales, Scedosporium, Histoplasma

Response 4: We recognize the importance of assessing whether samples were positive for other relevant fungal pathogens, such as Mucorales, Scedosporium, and Histoplasma. Details regarding this issue are provided in the Materials and Methods section.

Comments 5: This ms is not really exciting and it is not the first time that PCR is used to diagnose these two fungal infections.

Response 5: : While PCR-based assays have been previously used for the detection of A. fumigatus and P. jirovecii, our study represents the first implementation of a fully automated, multiplex real-time PCR assay for these two fungi on the Panther Fusion system. This integration offers a streamlined workflow that reduces manual handling and improves diagnostic efficiency. The novelty and clinical utility of our approach have been discussed in the manuscript.

Reviewer 2 Report

Dear author,

Your paper, entitled:  A Laboratory-Developed Assay for Simultaneous Detection of Aspergillus fumigatus and Pneumocystis jirovecii Pulmonary Pathogens, deals with very important subject as the design of a diagnostic test that would allow rapid and accurate diagnosis of invasive fungal infection which are constantly neglected even in the part of diagnostics. This is very interesting study and very nice written.

The most significant progress in the diagnosis of infectious diseases, fungal infections included, has been made with the introduction of molecular biology. Molecular detection and identification have a great importance in the diagnosis of infectious diseases, especially in cases where standard microbiological and mycological procedures have low sensitivity, or they are difficult for interpretation. Your manuscript is written clearly, highlighting the advantages and disadvantages, as well as the limitations in not linking the results to the clinical findings of the patients from whom BAL was sampled. Designed molecular tool can enhance the speed of diagnosis as well as simultaneous detection of Aspergillus fumigatus and Pneumocystis jirovecii as a sensitive and specific tool for the detection and identification of pathogens or opportunistic pathogens.

Only one suggestion from me is that you could highlight more the importance of multidisciplinary approach in diagnosis of pulmonary aspergillosis or PcP even with molecular diagnostics which detected and determined potential pathogen, but not infection.

line 242- change- corresèonding to corresponding

Author Response

Comments 1: Dear author,

Your paper, entitled: A Laboratory-Developed Assay for Simultaneous Detection of Aspergillus fumigatus and Pneumocystis jirovecii Pulmonary Pathogens, deals with very important subject as the design of a diagnostic test that would allow rapid and accurate diagnosis of invasive fungal infection which are constantly neglected even in the part of diagnostics. This is very interesting study and very nice written.

The most significant progress in the diagnosis of infectious diseases, fungal infections included, has been made with the introduction of molecular biology. Molecular detection and identification have a great importance in the diagnosis of infectious diseases, especially in cases where standard microbiological and mycological procedures have low sensitivity, or they are difficult for interpretation. Your manuscript is written clearly, highlighting the advantages and disadvantages, as well as the limitations in not linking the results to the clinical findings of the patients from whom BAL was sampled. Designed molecular tool can enhance the speed of diagnosis as well as simultaneous detection of Aspergillus fumigatus and Pneumocystis jirovecii as a sensitive and specific tool for the detection and identification of pathogens or opportunistic pathogens.

Only one suggestion from me is that you could highlight more the importance of multidisciplinary approach in diagnosis of pulmonary aspergillosis or PcP even with molecular diagnostics which detected and determined potential pathogen, but not infection.

Response 1: We fully agree with this important point. Molecular diagnostics, including our AF/PJ LDA, provide highly sensitive detection of fungal DNA but do not distinguish between colonization and active infection. We have expanded the discussion to underscore the need for a multidisciplinary approach, integrating clinical, radiological, and serological findings (e.g., BDG and GM assays) for a comprehensive diagnosis.

Comments 2: line 242- change- corresèonding to corresponding.

Response 2: Thank you for pointing this out. The typo has been corrected.